# Shoulder girdle injuries involving the medial clavicle differ from lateral clavicle injuries with a focus on concomitant injuries and management strategies: A retrospective study based on nationwide routine data

**M. Sinan Bakir**[1,2]*, **Jan Unterkofler**[2,3], **Alexander Hönning**[4], **Lyubomir Haralambiev**[1,2], **Simon Kim**[1], **Axel Ekkernkamp**[1,2], **Stefan Schulz-Drost**[2,5,6]

**1** Department of Trauma and Reconstructive Surgery and Rehabilitative Medicine, Medical University Greifswald, Greifswald, Mecklenburg-Vorpommern, Germany, **2** Department of Trauma Surgery and Orthopedics, BG Hospital Unfallkrankenhaus Berlin gGmbH, Berlin, Germany, **3** Department of Vascular Surgery, University Hospital RWTH Aachen, Aachen, Nordrhein-Westfalen, Germany, **4** Center of Clinical Science, BG Hospital Unfallkrankenhaus Berlin gGmbH, Berlin, Germany, **5** Department of Trauma and Orthopedic Surgery, University Hospital Erlangen, Erlangen, Bayern, Germany, **6** Department of Trauma, Orthopedic and Hand Surgery, Helios Hospital Schwerin, Schwerin, Mecklenburg-Vorpommern, Germany

* sinan.bakir@uni-greifswald.de

## Abstract

### Introduction

Although shoulder girdle injuries are frequent, those of the medial part are widely unexplored. Our aim is to improve the knowledge of this rare injury and its management in Germany by big data analysis.

### Methods

The data are based on ICD-10 codes of all German hospitals as provided by the German Federal Statistical Office. Based on the ICD-10 codes S42.01 (medial clavicle fracture, MCF) and S43.2 (sternoclavicular joint dislocation, SCJD), anonymized patient data from 2012 to 2014 were evaluated retrospectively for epidemiologic issues. We analyzed especially the concomitant injuries and therapy strategies.

### Results

A total of 114,003 cases with a clavicle involving shoulder girdle injury were identified with 12.5% of medial clavicle injuries (MCI). These were accompanied by concomitant injuries, most of which were thoracic and craniocerebral injuries as well as injuries at the shoulder/ upper arm. A significant difference between MCF and SCJD concerning concomitant injuries only appears for head injuries (p = 0.003). If MCI is the main diagnosis, soft tissue injuries typically occur as secondary diagnoses. The MCI are significantly more often associated with concomitant injuries (p < 0.001) for almost each anatomic region compared with lateral clavicle injuries (LCI). The main differences were found for thoracic and upper

**Data Availability Statement:** All raw data files are available from the Dryad Digital Repository (https://doi.org/10.5061/dryad.q5fk2d7).

**Funding:** The authors acknowledge support for the Article Processing Charge from the DFG (German Research Foundation, 393148499) and the Open Access Publication Fund of the University of Greifswald. The funder had no role in study design, data collection and analysis, decision to publish, or preparation of the manuscript.

**Competing interests:** The senior author has a consultant agreement with DePuySynthes and he is an advisory member of the AO TK Thoracic Surgery Expert Group (THEG). This does not alter our adherence to PLOS ONE policies on sharing data and materials. The other authors are not involved in any competing interests.

extremity injuries. Different treatment strategies were used, most frequently plate osteosynthesis in more than 50% of MCF cases. Surgery on SCJD was performed with K-wires, tension flange or absorbable materials, fewer by plate osteosynthesis.

## Conclusions

We proved that MCI are rare injuries, which might be why they are treated by inhomogeneous treatment strategies. No standard procedure has yet been established. MCI can occur in cases of severely injured patients, often associated with severe thoracic or other concomitant injuries. Therefore, MCI appear to be more complex than LCI. Further studies are required regarding the development of standard treatment strategy and representative clinical studies.

## Introduction

Clavicle injuries are a common entity of upper extremity injury [1]. A solitary clavicle fracture represents about 2.5–10.0% of all fractures [2–6]. However, most of these injuries are in the midshaft region rather than the medial clavicle [1, 7]. The dislocation of the sternoclavicular joint (SCJD) is often the result of a high velocity force experienced during trauma [5, 8]. Although numerous surgical treatments have been reported, epidemiologic data regarding such injuries in Germany is sparse; this applies particularly to concomitant injuries and therapeutic strategies that have so far been described only in case reports/series or in small studies with a minor level of evidence [9–18].

Nonetheless, medial clavicle injuries (MCI) are quite important, as this anatomic area is the most important articular joint connecting the upper extremity to the trunk. This has an important impact on shoulder-girdle kinematics and stability [19]. Moreover, posterior dislocation of the SCJ is associated with major concomitant complications, such as haematopneumothorax [20], tracheal injuries [21] and (neuro-)vascular compression problems [22].

Given this significance, we present an analysis of current epidemiologic data of medial clavicle fractures (MCF) and SCJD in Germany. The investigation focused primarily on the frequency and importance of MCI and comparing both variants of MCI with each other and with other clavicle injuries concerning concomitant injuries and the treatment strategies applied.

## Methods

The retrospective study was approved by the local ethics committee (Medical University Greifswald: BB 007/19). Since the data provided by the German Federal Statistical Office were purely retrospective and anonymized, no experiments on humans or animals had been done. Routine data based on the 10th revision of the International Statistical Classification of Diseases and Related Health Problems (ICD-10 codes) of all German hospitals discounting diagnosis-related groups (DRG) in the scope of application of § 1 of the German hospital finance law (KHEntgG) has been analyzed in detail [23, 24].

All patients released from in-patient settings (including those deceased) were included in this analysis. The ICD-10 codes S42.01, S42.02 and S42.03 (clavicle fracture medial, midshaft and lateral, respectively) in addition to S43.1 and S43.2 (acromioclavicular and sternoclavicular joint dislocations, respectively), including their combinations, were evaluated from 2012 to 2014. We extracted the data of MCI S42.01 and S43.2 from these five shoulder girdle injuries

involving the clavicle and focused on them. The lateral clavicle fracture and the acromioclavicular joint dislocation were summarized to the subgroup of lateral clavicle injuries (LCI) for comparison. The retrospective analysis addresses potential concomitant injuries and therapy strategies.

Concerning the concomitant injuries, we also distinguish between main and secondary diagnosis of MCI. Therefore, a patient with multiple injuries was counted once, since we analyzed each of the shoulder-girdle injuries relating to the clavicle for itself and the further injuries were counted as main or secondary diagnosis, vice versa, to the related injury. We focused on chapter XIX of the ICD-10 code, which contains "Injuries, poisoning and certain other consequences of external causes" for the analysis of concomitant injuries and excluded all posttraumatic conditions resulting in the analysis of diagnoses describing only primary injuries (S00-S99) [23]. Soft tissue damage is classified according to the Oestern and Tscherne classification in the ICD-10 code system, which is also used most often in the literature to describe soft tissue injuries in blunt trauma [25].

The therapies conducted were analyzed based on the German procedure classification ("Operationen- und Prozedurenschlüssel"; OPS code), which is the official classification for the encoding of operations, procedures and general medical measures [26]. The OPS is available in various versions and formats in German and is updated annually [26]. We focused on the OPS codes of the subdivided categories "operations" (5–01 . . .5–99) and "operations of movement organs" (5–78 . . .5–86), and the category "closed reduction and correction of deformities" (8–20. . .8–22). We analyzed these categories and further description of the targeted part "clavicle," "sternoclavicular joint" or "others" were inclusion criteria. Thus, the unrelated interventions were excluded. These were surgeries of concomitant injuries at other parts of the body, but related to the case due to an MCI as a main diagnosis. The relevant therapies were, without exception, part of the categories "operations at other bones" (5–78), "reduction of fracture and dislocation" (5–79), "open surgical and other joint operations" (5–80) and "closed reduction and correction of deformities" (8–20. . .8–22). The data were presented as summarized interventions of MCI as a main and secondary diagnosis. Therefore, all injuries with either surgical or nonsurgical treatment were included.

Statistical analysis was performed using the SPSS software (IBM, version 22, Champaign, IL). The association between type of fracture and frequency of surgery was tested by Pearson's chi-squared test and, in the case of low cell frequencies (i.e. single cell number < 5), via Fisher's exact test with an alpha level of 0.05. In some cases, confidence intervals were added. No alpha adjustment for multiple testing was conducted due to the explorative character of the analysis.

## Results

We reviewed a total of 114,003 patients who had a diagnosed clavicle injury (*Fig 1*). Of these, 12.5% are coded as MCI, n = 13,588 for medial clavicle fractures (S42.01; 11.9%) and n = 676 for SCJD (S43.2; 0.6%) of all clavicle injuries (*Fig 1*). This group is used for further investigation. The average patient age for SCJDs was 50.3 (±23.3) years and 47.7 (±22.8) years for MCFs; an average of 67.2% (CI 63.5–70.7%) of SCJD were attributed to males and 32.8% (29.3–36.5%) to females. The sex distribution of MCF was an average of 69.4% (CI 68.6–70.2%) in males and 30.6% (29.9–31.4%) in females.

### Concomitant injuries

MCI are associated with a significantly higher number of concomitant injuries at each anatomic region, apart from wrist and hand injuries, in contrast to LCI (*Fig 2*, p = 0.001–0.02).

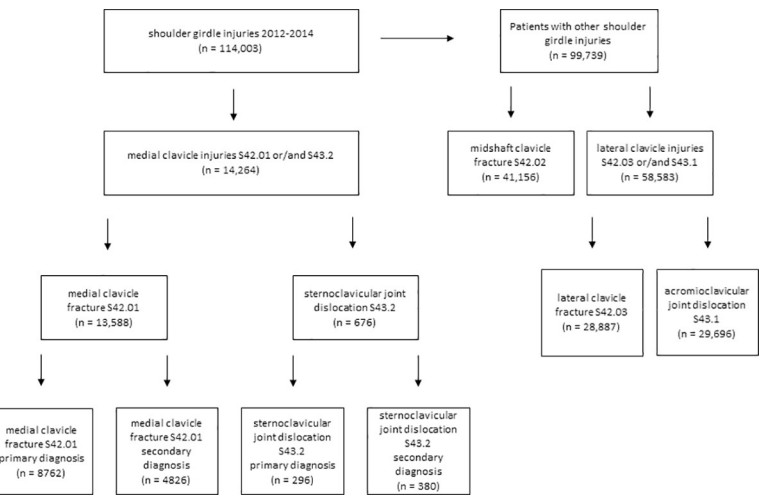

**Fig 1. Prism of distribution of shoulder girdle injuries relating to the clavicle, 2012–2014.** n = number of patients.

Concomitant injuries of the shoulder and upper arm are the most commonly affected part of the body in MCI and LCI, while the major differences between the medial and lateral shoulder girdle occur in this group and in concomitant thoracic injuries. The ratio of associated injuries overall to the number of diagnoses in the case of MCI is 20.4% higher than LCI. MCI have an average of 1.0 concomitant injuries per case, while LCI have a ratio of 0.8.

With a focus on MCI particularly, the most common concomitant injuries occur regarding the anatomical region affected at the shoulder/upper arm (SCJD 39.5%; MCF 42.2%), as thoracic injuries (SCJD 19.4%; MCF 19.6%) and as craniocerebral injuries (SCJD 14.1%; MCF 18.3%) (Fig 3). A significant difference between both MCI only appears for concomitant head injuries (p = 0.003).

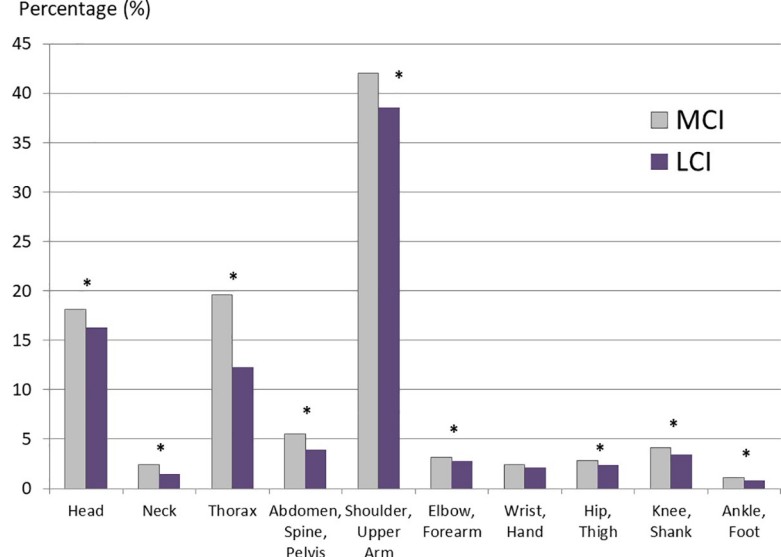

**Fig 2. Concomitant injuries of medial clavicle and lateral clavicle injuries categorized by the anatomical region.** MCI = medial clavicle injuries; LCI = lateral clavicle injuries; * = significant difference.

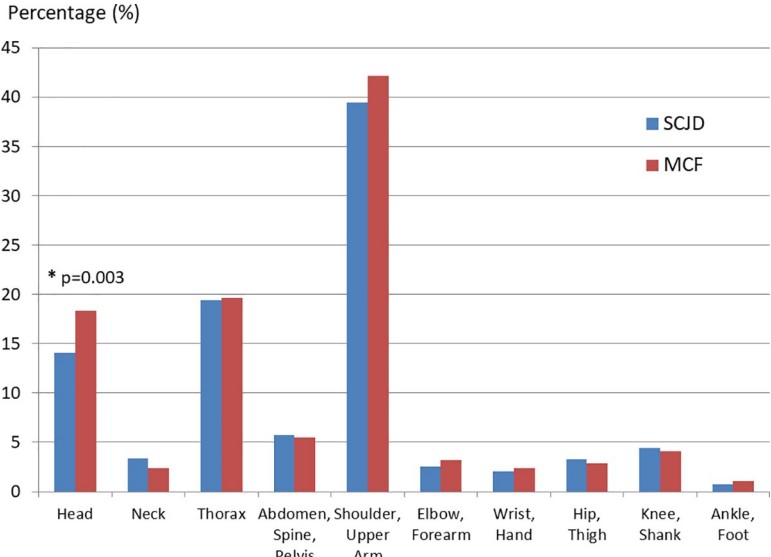

**Fig 3. Concomitant injuries of medial clavicle injuries categorized by the anatomical region affected.**
S43.2 = sternoclavicular joint dislocation; S42.01 = medial clavicle fracture; *p = p-value as level of significant difference.

Since every case of MCI as a secondary diagnosis has to be associated with another concomitant injury as a main diagnosis, we further focus on the opposite part. As a main diagnosis, a SCJD is associated with an average of 1.1 further concomitant injuries (*Fig 4*). The same applies to MCF, with a mean of 1.1 other diagnosis. These concomitant injuries are most frequently soft tissue injuries in both types of MCI.

The distribution is quite similar for each MCI with special regard to the specific thoracic concomitant injuries (*S1 Table*). The two most frequent are, in both cases, an associated serial

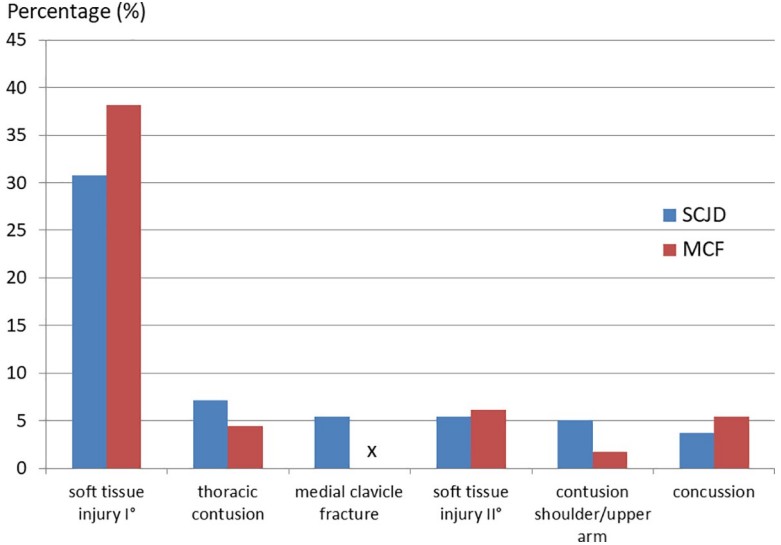

**Fig 4. Most common concomitant injuries of medial clavicle injuries as a main diagnosis.** Second diagnoses according to MCI as a main diagnosis presented as the five most common second diagnoses for both types of MCI each. S43.2 = sternoclavicular joint dislocation; S42.01 = medial clavicle fracture; x = non-valid, since main ≠ secondary diagnosis at the same time.

rib fracture with participation of four or more ribs (MCF 18.2% of all thoracic concomitant injuries; SCJD 14.5%) and a contusion of the thorax (MCF 15.2%; SCJD 17.6%). The thoracic first grade soft tissue injuries differ (MCF 8.3%; SCJD 1.5%) as do fractures of the sternum and first rib (MCF 4.0%; SCJD 9.9%).

## Therapy strategies

There was a significant difference between MCF and SCJD regarding the fundamental types of operations (*Fig 5*): here were significantly (p < 0.001) more removals of osteosynthetic material in the case of MCF (18.9%) compared to SCJD (7.5%) in relation to all interventions regarding SCJD and MCF. After exclusion of all irrelevant coded interventions–also excluding the removals of osteosynthetic material, coded by a related ICD-10 code of MCI in terms of a removal as a main diagnosis–there remained n = 9445 primary operations of MCI (*Fig 5*).

There were differences in treatment options within the MCI (*Table 1*). The ratio of primary surgical interventions to the number of injuries was 60.4% for SCJD, while 66.5% were treated surgically in the case of MCF. Both MCI were the domain of open surgical treatment (p < 0.001). While an open surgery was done in 92.2% of all MCF operations in contrast to 89.5% of all SCJD (p < 0.001), a closed procedure was performed in 7.8% of all MCF compared to 10.5% at SCJD (p = 0.049).

Heterogeneous treatment options were performed for both MCI in the case of an invasive surgical strategy with a reduction via osteosynthesis (*Fig 6*). Each specific type of osteosynthesis showed significant differences between SCJD and MCF (p < 0.001). While SCJD was a preserve of osteosynthesis via wire or tension flange, more than half of the MCF treated by osteosynthetic procedure received a (locking) plate osteosythesis.

## Discussion

The distribution of MCI is slightly different from the data published in the past: SCJD seem to be less frequent and MCF more frequent than assumed so far [3–5, 7]. Previous work with a large cohort concerning clavicle fractures was limited by not differentiating between the

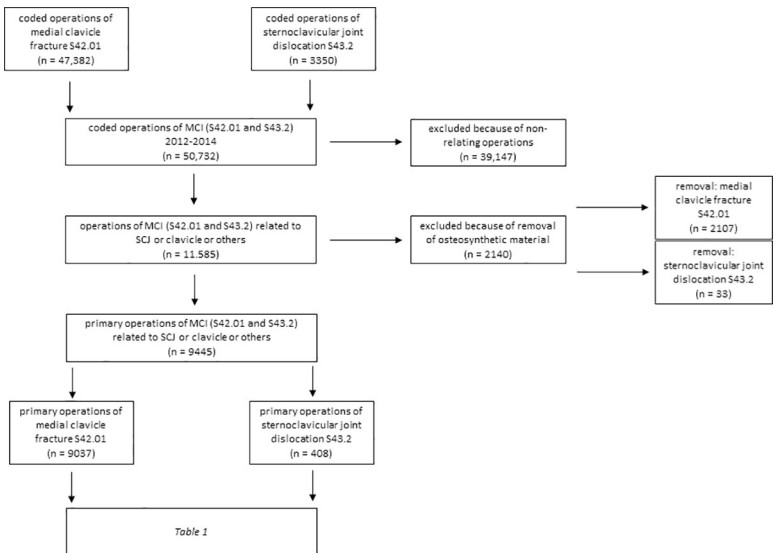

**Fig 5. Prism of medial clavicle injuries therapies coded by OPS code, 2012–2014.** MCI = medial clavicle injuries; SCJ = sternoclavicular joint; n = number of patients.

**Table 1. Number of particular primary interventions for medial clavicle injuries and the difference between the injury entities.**

| Category of treatment | Type of treatment | OPS code | S43.2 | | S42.01 | | Significance |
|---|---|---|---|---|---|---|---|
| | | | **n** | **% (CI)** | **n** | **% (CI)** | **p** |
| **Closed treatment** | closed reduction without osteosynthesis | 8–200 ff. | 37 | **9.1 (6.7–12.3)** | 65 | 0.7 (0.6–0.9) | < 0.001 |
| | closed reduction of fracture/ epiphyseal injury with osteosynthesis | 5–790 | 2 | 0.5 (0.1–17.7) | 643 | **7.1 (6.6–7.7)** | < 0.001 |
| | closed reduction of joint dislocation with osteosynthesis | 5-79a | 4 | **1.0 (0.4–2.5)** | 1 | <0.1 (0.0– < 0.1) | < 0.001 |
| **Open treatment** | open reduction of simple fracture at small bone | 5–795 | 34 | 8.3 (6.0–11.4) | 2703 | **30.0 (29.0–30.9)** | < 0.001 |
| | open reduction of multi-fragmentary fracture at small bone | 5–796 | 33 | 8.1 (5.8–11.1) | 5504 | **61.0 (59.9–61.9)** | < 0.001 |
| | open reduction of joint dislocation | 5-79b | 172 | **42.1 (37.5–47.0)** | 40 | 0.4 (0.3–0.6) | < 0.001 |
| | open joint surgery | 5–800 | 37 | **9.1 (6.7–12.3)** | 19 | 0.2 (0.1–0.3) | < 0.001 |
| | open surgery at joint cartilage or meniscus | 5–801 | 8 | **2.0 (1.0–3.8)** | 7 | <0.1 (<0.1–0.2) | < 0.001 |
| | open surgical refixation at capsular-ligamental system of other joints | 5–807 | 64 | **15.7 (12.5–19.5)** | 50 | 0.6 (0.4–0.7) | < 0.001 |
| **Others** | other joint surgeries | 5–809 | 17 | **4.1 (2.6–6.6)** | 5 | <0.1 (< 0.1–0.1) | < 0.001 |
| **Total** | | | 408 | 100.0 | 9037 | 100.0 | |

Only injuries related to sternoclavicular joint, clavicle or others were analyzed from 2012–2014 (n = 9445). Non-primary treatments, such as removal of osteosynthesis, have been excluded. The higher value is highlighted in bold. S43.2 = sternoclavicular joint dislocation; S42.01 = medial clavicle fracture; n = number of patients; % = percentage; CI = confidence interval; p = p-value as level of significance.

localizations of the fracture [27]. To the best of our knowledge, this is the epidemiologic study with the largest demographic sample analyzed in the literature. It might be that, based on the large cohort, these findings are more powerful than the past data. On the other hand, the advantage of the high number of cases could prove to be problematic. Some authors avoid, for example, specifying confidence intervals or presenting p-values in these cases as it makes relatively small differences significant and simulates a relationship that is purely statistically significant [28].

As we have shown, MCI are more frequently associated with concomitant injuries compared to LCI in all respects. The concomitant injuries of MCI appear predominantly in the upper half of the body, especially at the shoulder/upper arm, thorax and head. In comparison to recent work of clinical retrospective research, the high presence of concomitant injuries at these anatomic regions affected could be proved [15]. The concomitant craniocerebral injuries are significantly more in the case of MCF. This is quite unsuspected because of the assumption that a severe trauma impact leads more often to a SCJD instead of an MCF. However, in this study, we analyzed severely injured patients and monotraumatically injured ones. In order to deliver a more precise insight regarding the role of MCI in severely injured trauma patients with high impact trauma mechanism, further research with a focus on these circumstances is necessary. However, we confirm that MCI might be a hint of a severe thoracic trauma or trauma of the upper half of the body [29]. Therefore, patients with injuries to the medial part of the shoulder girdle should be examined with special care.

With a focus on specific concomitant injuries, there are differences for main and secondary diagnoses especially regarding thoracic injuries, which are often underestimated. The

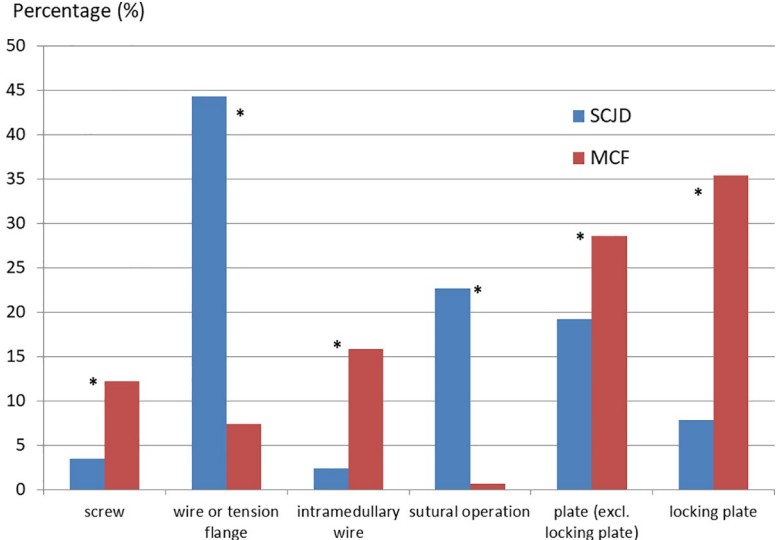

**Fig 6. Primary operations with osteosynthesis of medial clavicle injuries sorted for type of osteosynthesis.** Primary operations with osteosynthesis in the case of the coded diagnosis of a medial clavicle injury (S42.01 and S43.2) related to the sternoclavicular joint, clavicle or others sorted for the type of osteosynthesis. The number is presented as a percentage of all surgeries performed with osteosythesis. All types showed a significant difference between both diagnoses. S43.2 = sternoclavicular joint dislocation; S42.01 = medial clavicle fracture; * = p-value as level of significance with p < 0.001.

contemporaneous appearance of MCI and thoracic injuries is proved [30, 31]. While soft tissue injuries and costoclavicular injuries are found mostly in cases of MCF, the central localized sternal and first rib injuries occur predominantly with SCJD. This is presumed to be because of the close anatomic distance.

Since one concomitant injury per MCI case is only an average and there do not seem to be many at first glance, this rate is still 20% higher than with LCI and shows, therefore, the complexity of MCI. The frequently coded soft tissue injuries as a secondary diagnosis may be based on the attempt to enlarge the financial results. The minor proportions of several main diagnoses relating to MCI as a secondary diagnosis show their widely dispersed distribution.

There is currently no established protocol of care for sternal end clavicle injuries [32]. Although a new classification system and a proposal for a compulsory standard management for medial clavicle injuries have been published recently, these have not yet been evaluated or validated [15]. The heterogeneity of different treatment options of MCI was clearly identified in this study. Although the SCJD and the MCF are close to each other regarding their location, there are certain differences between the current treatment strategies of MCI.

In detail, the different corresponding anatomic structures of the SCJ and bony medial clavicle were referring to diverse treatments: Comprehensive operations relating to soft tissue, such as sutural surgery, were presented more often in SCJD, while most MCF were treated by plate osteosynthesis. In comparison with recent work of clinical retrospective research, there is a relevant difference to the amount of conservative/operative therapy relating to 68.4% conservatively treated medial injuries [15]. In an international comparison, even fewer surgically treated patients were shown with an amount of little or no MCI surgery in Belgium and Sweden [16, 18].

The heterogeneous treatment might be induced by different subtypes of MCI. A more precise classification of the medial injuries in terms of SCJD and medial clavicle fractures provided in recent work is not possible as this study is based only on routine data [15, 33].

Therefore, an attribution of the different treatment options to any type of classification is not possible. This is why a missing doublecheck referring to the correct radiologic location and a potential miscoding are also potential biases regarding this point [15]. Unfortunately, this problem remains unsolved in our study due to the underlying, purely retrospective anonymized data and, consequently, the lack of traceability of the individual cases. This is an important aspect, since a miscoding of medial clavicle fractures is a frequent issue, especially the mix-up between medial and middle third fractures [15]. Other potential bias that might increase the heterogeneity of treatment strategies and which could not have been prevented due to anonymized data are genetic disorders or oncologic patients with diseases affecting the musculoskeletal system. However, although these patients might be subject to adopted treatment, the number of these special cases is assumed not to be in relation to the total size of the cohort. We also admit that the encryption to OPS code is not unambiguous, since the operations are categorized only by major groups of surgical procedures.

Another possible confounder is coding regarding financial aspects, which could lead to an over-/under-coding of certain primary and secondary diagnoses [34, 35]. In addition, it is necessary to clarify the data from the falsification produced by coding the removals of osteosynthesis material, since there are 7.5% for SCJD and 18.9% for MCF of all relevant operations performed. This could also mix up the primary operated cases and the cases of a removal of osteosynthesis material and, therefore, lead to duplicate counting of the same patient over the years. This is attributed to an attendant and iterated coding of the original main diagnosis in the case of removal of material. The procedures and operations conducted and coded were analyzed. Because these were not the same as the number of cases, more than one procedure/operation per patient and case is possible. Several codes in the case of a complex operation are intended [26]. This could lead to a bias due to multi-coding.

A limitation in ICD and OPS coding is the code "others." Further differentiation is impossible in such cases. A direct conclusion from OPS code to the injured body part can usually be interpreted by the last number/letter of the code to avoid misinterpretation from OPS code to an actual false and non-corresponding injury. A bias or distortion resulting from changes in coding behavior or in the classification systems over the years are suspected to be marginal. The possibility of coding operations at muscles, tendons, fasciae and bursae (OPS 5–85) was missing for SCJ. Therefore, a small lack of OPS coding system might be possible.

Despite the limitations of this registry research and big data analysis in general, especially concerning the potential bias of an impossible and missing doublecheck for the exactness of coding, our analysis offers new aspects regarding MCIs in Germany. An important aspect that warrants further study is an analysis of combined shoulder girdle injuries, which occurred in these findings in contrast to single ones. An interesting area of investigation would be the distribution among different hospitals relating to the level of health care (basic, regular or maximum) and whether there is a correlation to the treatment strategies chosen. An SCJD is a particularly severe injury with potential concomitant or following complications and co-injuries, as shown. For this reason, the statutory accident insurance in Germany lists this injury in the register for very severe injury procedures with its own number, 7.2, which means a treatment in a specialized level one trauma center [36].

## Conclusion

Concomitant injuries are common in MCI. We attribute this fact to the high trauma force, which is often responsible for these entities [15, 27, 29, 37]. Concerning the spectrum of concomitant injuries, we demonstrated a relevant difference between medial and lateral clavicle injuries. MCF and SCJD differ only slightly in this regard. However, the variety of current

management strategies of MCI confirms the fact that there is currently no standard therapy of MCI. The status in Germany shows heterogeneous treatment options including operative and conservative therapy with a relevant tendency to surgery.

## Supporting information

**S1 Table. Concomitant thoracic injuries of medial clavicle injuries.** Concomitant thoracic injuries of sternoclavicular joint dislocations (S43.2) and of medial clavicle fractures (S42.01) are presented as an absolute and relative number in relation to all concomitant thoracic injuries categorized by the specific thoracic injury from 2012 to 2014. ICD = code of the diagnosis by ICD-10; SD = secondary diagnosis; MD = main diagnosis; SD+MD = all diagnoses (secondary and main diagnosis); S43.2 = sternoclavicular joint dislocation; S42.01 = medial clavicle fracture; n = absolute number of cases; % = relative amount in relation to all concomitant thoracic injuries of the respective medial clavicle injury.
(XLSX)

## Author Contributions

**Conceptualization:** M. Sinan Bakir, Jan Unterkofler, Stefan Schulz-Drost.

**Data curation:** M. Sinan Bakir, Jan Unterkofler, Stefan Schulz-Drost.

**Formal analysis:** M. Sinan Bakir, Alexander Hönning.

**Funding acquisition:** M. Sinan Bakir.

**Investigation:** M. Sinan Bakir, Stefan Schulz-Drost.

**Methodology:** M. Sinan Bakir, Jan Unterkofler, Alexander Hönning, Stefan Schulz-Drost.

**Project administration:** M. Sinan Bakir, Axel Ekkernkamp, Stefan Schulz-Drost.

**Resources:** M. Sinan Bakir, Lyubomir Haralambiev, Axel Ekkernkamp, Stefan Schulz-Drost.

**Software:** M. Sinan Bakir, Alexander Hönning, Simon Kim.

**Supervision:** M. Sinan Bakir, Lyubomir Haralambiev, Axel Ekkernkamp, Stefan Schulz-Drost.

**Validation:** M. Sinan Bakir, Lyubomir Haralambiev, Simon Kim, Axel Ekkernkamp.

**Visualization:** M. Sinan Bakir.

**Writing – original draft:** M. Sinan Bakir, Jan Unterkofler, Alexander Hönning, Lyubomir Haralambiev, Simon Kim, Stefan Schulz-Drost.

**Writing – review & editing:** M. Sinan Bakir, Lyubomir Haralambiev, Simon Kim, Axel Ekkernkamp, Stefan Schulz-Drost.

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
