## [Decision Letter · Decision Letter 0]

30 Aug 2019

[EXSCINDED]

PONE-D-19-21318

Shoulder girdle injuries involving the medial clavicle differ from lateral clavicle injuries with focus on concomitant injuries and management strategies: A retrospective study based on nationwide routine data

PLOS ONE

Dear Dr. med. Bakir,

Thank you for submitting your manuscript to PLOS ONE. After careful consideration, we feel that it has merit but does not fully meet PLOS ONE’s publication criteria as it currently stands. Therefore, we invite you to submit a revised version of the manuscript that addresses the points raised during the review process.

The authors put many hours in collecting data and preparing this interesting manuscript. The recommended therapeutic procedures for MCF as well as for SCJD are really very heterogenous, which underlines the importance of this study. However, in the present form it cannot be accepted for publication in POLS ONE. Please refer to the careful done reviwes for improvement. As pointed out by both reviewers, I believe that a native speaker could be of great linguistic help.

We would appreciate receiving your revised manuscript by Oct 14 2019 11:59PM. To enhance the reproducibility of your results, we recommend that if applicable you deposit your laboratory protocols in protocols.io, where a protocol can be assigned its own identifier (DOI) such that it can be cited independently in the future. For instructions see: http://journals.plos.org/plosone/s/submission-guidelines#loc-laboratory-protocols

We look forward to receiving your revised manuscript.

Kind regards,

Hans-Peter Simmen, M.D., Professor of Surgery

Academic Editor

PLOS ONE

Journal Requirements:

2. In ethics statement in the manuscript and in the online submission form, please provide additional information about the database used in your retrospective study. Specifically, please ensure that you have discussed whether all data were fully anonymized before you accessed them and/or whether the IRB or ethics committee waived the requirement for informed consent. If patients provided informed written consent to have their data used in research, please include this information.

Reviewers' comments:

Reviewer's Responses to Questions

**Comments to the Author**

1. Is the manuscript technically sound, and do the data support the conclusions?

Reviewer #1: Partly

Reviewer #2: Partly

2. Has the statistical analysis been performed appropriately and rigorously? 

Reviewer #1: Yes

Reviewer #2: I Don't Know

3. Have the authors made all data underlying the findings in their manuscript fully available?

Reviewer #1: Yes

Reviewer #2: Yes

4. Is the manuscript presented in an intelligible fashion and written in standard English?

Reviewer #1: Yes

Reviewer #2: No

5. Review Comments to the Author

Reviewer #1: Dear Autors

Thank you for submitting your paper to this journal.

You submit a scientific paper focusing on the frequency of medial clavicule injuries including medial clavicule fractures as well as dislocations of the sternoclaviular joint. Their cohort included a high number of patients (13588 MCF and 676 SCJD) and therefore the study could be very interresting and could bring up relevant differences between MCF and SCJD as well as between MCI and LCI. The severe problem with this data analysis is, that it is based on numbers related to ICD coding and OPS coding. We all know, that this may significantly falsify data, as coding is sometimes driven by financial and economic interrests. Although some conclusions seem to be logical, I would not take the data as realistic and meaningful. Over and/or undercoding as well as multicoding should be ruled out by looking at the charts and x-rays of the patients to avoid misinterpretation.

If the data are correct and doublechecked, the paper would contribute to a better understanding of shoulder girdle injuries. The paper is well written, the citations are correct and I only would suggest a few revisions.

Page 4 line 55: Change to: This has significant impact on Shoulder girdle kinematics as well as stability.

Page 4 line 60: The investigation focused primarily on the frequency and importance of MCI and comparing both variants as well as comparing MCI with other clavicle injuries concerning concomitant injuries and applied treatment strategies.

Page 5 line 80: chapter XIX is not clear for all readers of the journal.

Page 7 line 102: I would suggest to skip Fig I. The clarification in the text seems to be enough.

Page 8 line 129 to 149: this paragraph could possibly improve by rewriting and shortening by focusing on the relevant data. Why is it so important if the diagnosis is coded as the primary or secundary diagnosis keeping the uncertainty of coding in mind. Under this perspective also figure 5a and 5b could be revised.

Regarding the treatment options it seems very difficult to draw or follow relevant conclusions, as there is no detailed information on the type, classification and pattern of the injury. This should also be provided after analysing the charts and x-rays.

Concluding I want to emphysize, that in my opinion this paper would gain a lot of significance by correction of the data by checking the charts and the x-rays and improve the quality of the basic data to reduce this uncertainty.

Reviewer #2: The authors address the topic of medial clavicle injuries by using a nationwide database.

Abstract

Introduction

I have a little problem with the phrase “general idea”. Please rephrase this sentence into the hypothesis/research question/or the aim of this study.

Methods

Based on ICD-10 codes patients with medial clavicle fractures or SC joint dislocation were evaluated. This methodology could be adequate depending on the research question.

Results

The results section of the abstract need major revision: I recommend English correction by a native speaker. “A significant difference between the both MCI only appears for concomitant head injuries (p=0.003)” This sentence appears incomplete: what groups were compared?

Conclusion

Major language correction, conclusion of MCI being rare with inhomogeneous treatment strategies

Manuscript

Intoduction

The first sentence states clavicle fractures to be a common complication of upper extremity injury. We believe that these fractures are not common complication of other injuries, but are common injuries by themselves. Further, the authors state epidemiologic data regarding such injuries in Germany were sparse and cite 6 articles. This seems inconsistence.

“MCI are quite important as they are the only articular joint of the upper extremity to the trunk” Technically speaking, the scapula-thoracic also serves as an articular joint of the upper extremity to the trunk. Please correct the sentence (e.g. one of the important….)

“This has significant impact on…”, “significant” usually followed by a p-value / a comparison. Please rephrase

I have problems with the last sentence of the Introduction: The investigation focused primarily on the role of MCI in their comparison -> to what? And in comparison to other clavicle injuries concerning concomitant injuries …. This phrase is hard to read. The reader would benefit from substantial revisions with the help of native speaker

Methods

According to the methodology of this article, the authors only evaluated ICD-10 diagnosis an OPS codes.

Please include a definition of shoulder girdle injuries relating to the clavicle

The time frame 2012-2014 should be included in the methods section

Please include inclusion and exclusion criteria: where only surgical treated injuries included, or all injuries (non-surgical treatment). What about genetic disorder, or oncological patients with diseases affecting the musculo-skeletal system: these patients might be subject to adopted treatment that might increase heterogeneity of treatment strategies.

Are multiple injured patients calculated multiple times? E.g. Patient with MCF and, pneumothorax and fracture of the ankle. How is this patient being handled?

Please define soft tissue injury

How where patients handled that were operated several different times due to MCI / MCF?

Results

Figures: Generally, please replace the ICD-10 code with the wording of the injury, eventhough it has been stated in the description, I believe the reader would benefit from stating the text in the graph, rather in the description. X = non valid (where there missing data? NAs?)

How many patients had only the injury to the medial part of the clavicle, or is there always a concomitant injury?

Line 129: Concerning the concomitant injuries in particular… This belongs to the methods section

Line 143: There are also some differences for main as well as for secondary diagnosis with special regard to…. -> Please revise this sentence; don’t use empty phrases such as “There are also some differences”

It appears the results section to be a collection of Figure and Table descriptions…

Discussion

Major English corrections needed

Line 186-187: The authors state that large cohort analysis are more precise: We highly disagree with this statement. The problem with big data analysis, especially when comparing groups has been discussed several times. Further, this study lacks substantial elements of epidemiologic studies: Table 1 approach, patients demographics etc. Citing an article regarding head injuries the authors state “specifying confidence intervals in these cases” to be avoided “as it makes relatively small differences significant and simulates a relationship that is purely statistically significant”. First, this statement holds true for all statistical tests, and tor the p-value. The alpha is arbitrary set at 0.05, and the statistical tests simulate a purely statistical significance. Second, the CI gives way more information than the p-value and should always be preferred.

Line 190: “Another potential bias…” what has been done to reduce this problem?

The discussion part appears disorganized. Usually, the main results are discussed point by point and one section is dedicated to strengths and limitations. The limitations of this study are discussed in lines 191ff, 228, 247ff, 255. Apparently the author included a “limitation-sentence” at the end of each paragraph. I would not recommend doing so: First, usually Strengths and limitations have one section; Second, with the authors approach, each paragraph/section loses weight, when ended by a “limitation-sentence”

Conclusion:

Line 266: “… the high trauma force which is often responsible for this entities”. I don’t find data in the authors article to support this conclusion. How was the trauma force measured/calculated? Have these data been included?

“on the other hand” empty phrase, discard

6. PLOS authors have the option to publish the peer review history of their article (what does this mean?). If published, this will include your full peer review and any attached files.

Reviewer #1: No

Reviewer #2: No

---

## [Author Response · Author response to Decision Letter 0]

11 Oct 2019

Reviewer #1: Dear Autors

Thank you for submitting your paper to this journal.

You submit a scientific paper focusing on the frequency of medial clavicule injuries including medial clavicule fractures as well as dislocations of the sternoclaviular joint. Their cohort included a high number of patients (13588 MCF and 676 SCJD) and therefore the study could be very interresting and could bring up relevant differences between MCF and SCJD as well as between MCI and LCI. The severe problem with this data analysis is, that it is based on numbers related to ICD coding and OPS coding. We all know, that this may significantly falsify data, as coding is sometimes driven by financial and economic interrests. Although some conclusions seem to be logical, I would not take the data as realistic and meaningful. Over and/or undercoding as well as multicoding should be ruled out by looking at the charts and x-rays of the patients to avoid misinterpretation.

If the data are correct and doublechecked, the paper would contribute to a better understanding of shoulder girdle injuries. The paper is well written, the citations are correct and I only would suggest a few revisions.

- Thank you for your kind evaluation. Of course, the problem of miscoding is enormously important. Especially in the analysis of big data, systematic coding bias can lead to wrong conclusions. We described this aspect in our limitations. Although individual cases of miscoding could be a negligible mistake regarding the large study cohort, a potential systematic bias is an important factor. Therefore, we have elaborated this point in the discussion.

Due to the anonymized data underlying this study and the purely retrospective examination, a doublecheck by controlling the x-rays is unfortunately not possible. However, as a conclusion, this is an important study in the future and case-based follow-up studies on this topic should result from this work to avoid miscoding bias and to measure this effect.

Page 4 line 55: Change to: This has significant impact on Shoulder girdle kinematics as well as stability.

Page 4 line 60: The investigation focused primarily on the frequency and importance of MCI and comparing both variants as well as comparing MCI with other clavicle injuries concerning concomitant injuries and applied treatment strategies.

Page 5 line 80: chapter XIX is not clear for all readers of the journal. 

- Thank you for your suggestions. We have implemented your suggestions and optimized the formulations.

Page 7 line 102: I would suggest to skip Fig I. The clarification in the text seems to be enough.

- Thank you for this advice. We skipped this figure to avoid duplication.

Page 8 line 129 to 149: this paragraph could possibly improve by rewriting and shortening by focusing on the relevant data. Why is it so important if the diagnosis is coded as the primary or secundary diagnosis keeping the uncertainty of coding in mind. Under this perspective also figure 5a and 5b could be revised.

- We have implemented this note and considered the MCI as a main diagnosis in the current version of the manuscript and the concomitant injuries as a secondary diagnosis only. Consequently, we deleted Figure 5b without replacement.

Regarding the treatment options it seems very difficult to draw or follow relevant conclusions, as there is no detailed information on the type, classification and pattern of the injury. This should also be provided after analysing the charts and x-rays.

Concluding I want to emphysize, that in my opinion this paper would gain a lot of significance by correction of the data by checking the charts and the x-rays and improve the quality of the basic data to reduce this uncertainty.

- In summary, we thank you for the constructive criticism and valuable comments that have improved our work. The main suggestion to control the cases radiologically would of course increase the quality and validity of the work. However, this was and is not possible due to the existing anonymized routine data and was not the aim of this work. The high number of cases analyzed as done in this cohort analysis could not have been achieved in the past, which is proved by previous work on this subject with a significantly smaller number of patients, but radiologically controlled data. Therefore, we believe that despite this limitation, this work has its value because of its large sample size and adds new aspects to the topic.

Nonetheless, we are already in the last steps of planning a prospective, case-controlled follow-up study, in which we follow a similar approach to image-controlled data.

Reviewer #2: The authors address the topic of medial clavicle injuries by using a nationwide database.

Abstract

Introduction

I have a little problem with the phrase “general idea”. Please rephrase this sentence into the hypothesis/research question/or the aim of this study.

Methods

Based on ICD-10 codes patients with medial clavicle fractures or SC joint dislocation were evaluated. This methodology could be adequate depending on the research question.

Results

The results section of the abstract need major revision: I recommend English correction by a native speaker. “A significant difference between the both MCI only appears for concomitant head injuries (p=0.003)” This sentence appears incomplete: what groups were compared?

Conclusion

Major language correction, conclusion of MCI being rare with inhomogeneous treatment strategies

- Thank you for your feedback. We followed your suggestions, reformulated the abstract and it was edited by a native speaker.

Manuscript

Intoduction

The first sentence states clavicle fractures to be a common complication of upper extremity injury. We believe that these fractures are not common complication of other injuries, but are common injuries by themselves. Further, the authors state epidemiologic data regarding such injuries in Germany were sparse and cite 6 articles. This seems inconsistence.

- We have worked on the ambiguity in the formulations and thank you for the advices. The six quoted articles confirm our statement, which we have related to both sentences more clearly: the topic has been described only in case reports / series or small studies with minor level of evidence in the past. 

“MCI are quite important as they are the only articular joint of the upper extremity to the trunk” Technically speaking, the scapula-thoracic also serves as an articular joint of the upper extremity to the trunk. Please correct the sentence (e.g. one of the important….)

“This has significant impact on…”, “significant” usually followed by a p-value / a comparison. Please rephrase

I have problems with the last sentence of the Introduction: The investigation focused primarily on the role of MCI in their comparison -> to what? And in comparison to other clavicle injuries concerning concomitant injuries …. This phrase is hard to read. The reader would benefit from substantial revisions with the help of native speaker

- We clarified the noted parts and revised them.

Methods

According to the methodology of this article, the authors only evaluated ICD-10 diagnosis an OPS codes.

Please include a definition of shoulder girdle injuries relating to the clavicle

The time frame 2012-2014 should be included in the methods section

- We presented the two aspects addressed in the method section of the first submitted draft (line 81-84). Since this was obviously not sufficiently highlighted and to clarify the definition, we have adapted this part.

Please include inclusion and exclusion criteria: where only surgical treated injuries included, or all injuries (non-surgical treatment). What about genetic disorder, or oncological patients with diseases affecting the musculo-skeletal system: these patients might be subject to adopted treatment that might increase heterogeneity of treatment strategies.

Are multiple injured patients calculated multiple times? E.g. Patient with MCF and, pneumothorax and fracture of the ankle. How is this patient being handled?

- We have re-sorted the inclusion and exclusion criteria, which were already mentioned in the first draft, to make them more recognizable (line 81-91, 106-114). We have implemented this for ICD 10 codes and OPS codes.

We gratefully took note of the aspect of patients with genetic disorder or oncological patients and included them in the discussion (line 251-255).

Please define soft tissue injury

- We have added a corresponding explanation in the methods section of the manuscript. In the case of the corresponding coding, the classification of the soft tissue damage of Tscherne and Oestern was used.

How where patients handled that were operated several different times due to MCI / MCF?

- Multiple coding is an important and potential confounding factor. Therefore, this problem has been addressed in our discussion (line 267-277).

Results

Figures: Generally, please replace the ICD-10 code with the wording of the injury, eventhough it has been stated in the description, I believe the reader would benefit from stating the text in the graph, rather in the description. X = non valid (where there missing data? NAs?)

- Thank you for your advice. We improved the comprehensibility and replaced the name of the ICD-10 code with the abbreviation of the injury.

The non-valid data refers to the fact that an injury, in the case mentioned the MCF, can not be both main and secondary diagnosis at the same time. We have completed and executed this in the picture caption.

How many patients had only the injury to the medial part of the clavicle, or is there always a concomitant injury?

- As described in Figure 1 (former Figure 2), 13,588 MCF occurred: 8762 as a main diagnosis, 4826 as a secondary diagnosis. n = 16 out of 296 patients with SCJD as a major diagnosis had a MCF as a secondary diagnosis. Therefore, MCF is present as a single diagnosis as well. MCF is not always present as a concomitant injury.

Line 129: Concerning the concomitant injuries in particular… This belongs to the methods section

- Due to your advice, we have moved the paragraph mentioned to the methods section (line 86-89).

Line 143: There are also some differences for main as well as for secondary diagnosis with special regard to…. -> Please revise this sentence; don’t use empty phrases such as “There are also some differences”

- We optimized the sentence accordingly, thank you very much for your advice.

It appears the results section to be a collection of Figure and Table descriptions…

- Thank you for your comment. Since, according to the journal's guidelines, the submission is prescribed like this, the description of the table and the figures have to be integrated into the text as done. Since the results are already presented via figures and tables, we passed on a detailed description of the results in the text of the result section, since we wanted to avoid duplication of data presentation.

Discussion

Major English corrections needed

Line 186-187: The authors state that large cohort analysis are more precise: We highly disagree with this statement. The problem with big data analysis, especially when comparing groups has been discussed several times. Further, this study lacks substantial elements of epidemiologic studies: Table 1 approach, patients demographics etc. Citing an article regarding head injuries the authors state “specifying confidence intervals in these cases” to be avoided “as it makes relatively small differences significant and simulates a relationship that is purely statistically significant”. First, this statement holds true for all statistical tests, and tor the p-value. The alpha is arbitrary set at 0.05, and the statistical tests simulate a purely statistical significance. Second, the CI gives way more information than the p-value and should always be preferred.

- Thank you for your detailed comment on this topic. We discuss this problem of the big data analysis in the relevant section of the discussion. We mention the problem of a high number of cases in terms of the significance of statistical tests. Confidence intervals have been added to Table 1. We also supplemented epidemiological data regarding age and sex distribution for a better overview on patients’ demographics (line 128-130).

Line 190: “Another potential bias…” what has been done to reduce this problem?

- Unfortunately, this problem remains unsolved due to the underlying, purely retrospective anonymized data and consequently the lack of traceability of the individual cases. We added this explanation to this part of the discussion.

The discussion part appears disorganized. Usually, the main results are discussed point by point and one section is dedicated to strengths and limitations. The limitations of this study are discussed in lines 191ff, 228, 247ff, 255. Apparently the author included a “limitation-sentence” at the end of each paragraph. I would not recommend doing so: First, usually Strengths and limitations have one section; Second, with the authors approach, each paragraph/section loses weight, when ended by a “limitation-sentence”

- Many thanks for this conclusive proposal, which we have gladly implemented in this form. We revised the part of the discussion and changed its structure.

Conclusion:

Line 266: “… the high trauma force which is often responsible for this entities”. I don’t find data in the authors article to support this conclusion. How was the trauma force measured/calculated? Have these data been included?

“on the other hand” empty phrase, discard

- We gratefully accepted the comment to our ambiguous phrase and clearly marked the corresponding line as our hypothesis and conclusion. An analysis of the trauma mechanism was not part of this study. In addition, we have also revised this section linguistically.

- In conclusion, we would like to thank you for the constructive evaluation. We have realized the main suggestion of a linguistic revision by a native speaker. We believe that, in spite of the remaining methodological optimization suggestions, this work is nevertheless worth publishing, since it contains aspects that have received little attention in the past regarding MCI. These aspects have been examined in this study analyzing a large cohort.

---

## [Editor Report · Decision Letter 1]

14 Oct 2019

Shoulder girdle injuries involving the medial clavicle differ from lateral clavicle injuries with a focus on concomitant injuries and management strategies: A retrospective study based on nationwide routine data

PONE-D-19-21318R1

Dear Dr. Bakir,

We are pleased to inform you that your manuscript has been judged scientifically suitable for publication and will be formally accepted for publication once it complies with all outstanding technical requirements.

With kind regards,

Hans-Peter Simmen, M.D., Professor of Surgery

Academic Editor

PLOS ONE

Additional Editor Comments (optional):

Thank you for answering to all reviewers comments.
---

## [Editor Report · Acceptance letter]

18 Oct 2019

PONE-D-19-21318R1 

Shoulder girdle injuries involving the medial clavicle differ from lateral clavicle injuries with a focus on concomitant injuries and management strategies: A retrospective study based on nationwide routine data 

Dear Dr. Bakir:

I am pleased to inform you that your manuscript has been deemed suitable for publication in PLOS ONE. Congratulations! Your manuscript is now with our production department. 

With kind regards,

on behalf of

Dr. Hans-Peter Simmen 

Academic Editor

PLOS ONE